# High-efficiency magnetic refrigeration using holmium

Noriki Terada [1✉] & Hiroaki Mamiya [1✉]

Magnetic refrigeration (MR) is a method of cooling matter using a magnetic field. Traditionally, it has been studied for use in refrigeration near room temperature; however, recently MR research has also focused on a target temperature as low as 20 K for hydrogen liquefaction. Most research to date has employed high magnetic fields (at least 5 T) to obtain a large entropy change, which requires a superconducting magnet and, therefore, incurs a large energy cost. Here we propose an alternative highly efficient cooling technique in which small magnetic field changes, $\Delta \mu_0 H \leq 0.4$ T, can obtain a cooling efficiency of $-\Delta S_M / \Delta \mu_0 H = 32$ J kg$^{-1}$K$^{-1}$T$^{-1}$, which is one order of magnitude higher than what has been achieved using typical magnetocaloric materials. Our method uses holmium, which exhibits a steep magnetization change with varying temperature and magnetic field. The proposed technique can be implemented using permanent magnets, making it a suitable alternative to conventional gas compression–based cooling for hydrogen liquefaction.

[1] National Institute for Materials Science, Tsukuba, Ibaraki, Japan. ✉email: terada.noriki@nims.go.jp; mamiya.hiroaki@nims.go.jp

Energy storage is a key issue that needs to be addressed for widespread adoption of renewable energy resources. In this regard, liquid hydrogen is widely expected to serve as medium for storing renewable energy resources; however, at present, the gas (helium or hydrogen) compression cycle cooling technique widely used for liquefaction of hydrogen suffers from high operational costs[1].

Magnetic refrigeration (MR) is an alternative refrigeration technique to gas compression[2]. MR, which is a method for cooling matter using a magnetic field, was discovered more than 100 years ago[3,4]. For the past 50 years, MR has been intensively studied for applications involving near room temperature refrigeration[5]. More recently, researchers in the field of MR have started focusing on lower target temperatures—especially, for the purpose of hydrogen liquefaction, which requires cooling hydrogen gas to a condensation temperature of 20 K[2]. MR cools matter by using magnetic entropy changes that occur when a magnetic field is applied, a phenomenon termed as magnetocaloric effect (MCE)[6]. MCE, also known as adiabatic diamagnetization among low temperature physicists[7], is quantified by the thermodynamic formula based on the Maxwell relation:

$$\Delta S_M = \mu_0 \int_{\mu_0 H_1}^{\mu_0 H_2} \left( \frac{\partial M(T,H)}{\partial T} \right)_H dH \qquad (1)$$

where $\Delta S_M$ is the magnetic entropy change during magnetic field ($H$) change from $H_1$ to $H_2$ and $M$ is the magnetization. The adiabatic temperature change defined by $\Delta T \equiv \Delta S_M / C$ ($C$ is the specific heat) is also an important quantity to evaluate MR efficiency of a material.

A large $\Delta S_M$ translates to a large change in magnetization with change in temperature. Thus, usually ferromagnetic (FM) materials at their Curie temperature $T_C$ are considered for MR applications. Figure 1a shows a simple MR cycle A → B → C → A of a model FM material. The largest reduction of magnetic entropy is achieved with magnetic fields where the Zeeman energy is comparable to the thermal energy at $T_C$. The required magnetic field is more than several Teslas in the temperature range between the hydrogen (20.3 K) and nitrogen (77 K) liquefaction temperatures. As a result, the use of stronger magnetic fields, with larger $\Delta S_M$ in FM materials has been considered more suitable for hydrogen liquefaction, despite the energy cost of generating the magnetic field increasing steeply in proportion to the square of the field amplitude.

Several candidate MR materials have been proposed for hydrogen liquefaction. For example, HoB$_2$ has been recently reported to have the largest MCE $\Delta S_M = 40.1$ J kg$^{-1}$ K$^{-1}$ among MR materials with a phase transition near the hydrogen liquefaction temperature[8]. $R$B$_4$ ($R$ = Dy, Ho) have also been recently reported to show a large magnetic entropy change achieved by a strong coupling between spin and quadrupolar degrees of freedom[9]. However, high magnetic fields of at least 2 T are necessary to obtain a large entropy change, which requires use of a superconducting magnet and, consequently, increases energy costs significantly.

Magnetic materials exhibiting a large magnetization jump (metamagnetization) show a large magnetic entropy change $\Delta S_M$ in a narrow magnetic field range. The magnetization change achieved in a minor cycle, A′ → B′ → C′ → A′, using a tiny field change can be almost identical to that obtained in the major A → B → C → A cycle (shown in Fig. 1b) and can cause a sufficient MR effect. There are some examples showing a large magnetization change in a narrow magnetic field range; for example, rare-earth inter metallic ErCo$_2$[10] and oxide ceramic (Sm$_{0.5}$Gd$_{0.5}$)$_{0.55}$Sr$_{0.45}$MnO$_3$[11]. In both cases, however, it is necessary to apply a magnetic field higher than 3 T to change the magnetization. In this study, we show that the rare-earth single metal holmium (Ho) exhibits a large magnetization change in a magnetic field lower than 1 T for temperatures ranging from 20 to 50 K. The critical field, $\mu_0 H_c$, which results in a large magnetization change is much smaller than that generated by a superconducting magnet. For example, $\mu_0 H_c = 0.2$ T at $T = 20$ K, and $\mu_0 H_c = 1.0$ T at $T = 50$ K. A large change can also be driven by a small variation in magnetic field $\Delta \mu_0 H \leq \sim 0.4$ T at each $\mu_0 H_c$. Consequently, a large MCE can be driven by a small variation in the magnetic field, allowing for MR with considerably reduced energy costs.

## Results and discussion

**Magnetic entropy change.** Ho is known as an antiferromagnetic (AFM) material with a helicoidal spin structure[12–15]. The phase transition from the paramagnetic to AFM helicoidal phases occurs at $T = 132$ K, which is followed by appearance of the ferromagnetic phase below $T = 20$ K in zero magnetic field[12]. The MCE of Ho in a strong magnetic field (6 T) has been reported previously[16]. However, in some AFM cases, such as Ho, a large and steep magnetization change, caused by the metamagnetic transition, occurs even when a small magnetic field is applied.

Such steep and large magnetization changes are observed in Ho single crystals when a magnetic field is applied along the

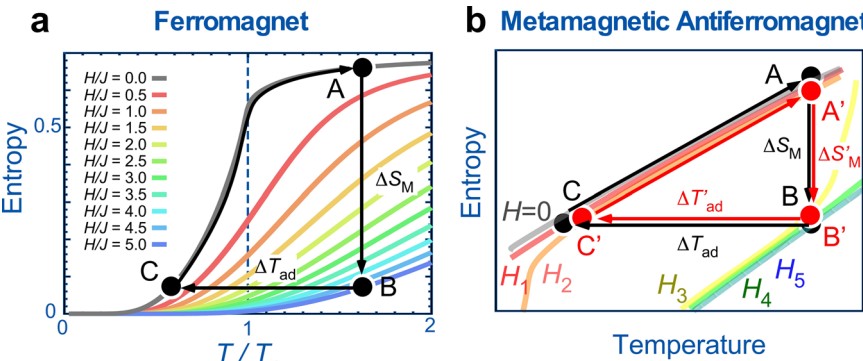

**Fig. 1 Comparison between conventional ferromagnet and antiferromagnet with a metamagnetic phase transition in entropy−temperature diagram. a** Thermal variation of magnetic entropy change in typical magnetic fields ($H$) from $H/J = 0$ to $H/J = 5.0$ (where $J$ is the ferromagnetic exchange constant) for ferromagnetic cases simulated in a previous theoretical study in ref. [30]. The temperature is normalized to the Curie temperature $T_C$. **b** Schematic illustration of magnetic refrigeration (MR) cycle for the case of antiferromagnetic (AFM) material that shows a metamagnetic phase transition. Magnetic fields are represented by zero field $\mu_0 H = 0$ and finite fields $H_1 < H_2 < H_3 < H_4 < H_5$. For the AFM case in **b**, the metamagnetic phase transition occurs over a small magnetic field change, $\Delta H = |H_2 - H_3|$, during the MR cycle A′ → B′ → C′ → A′.

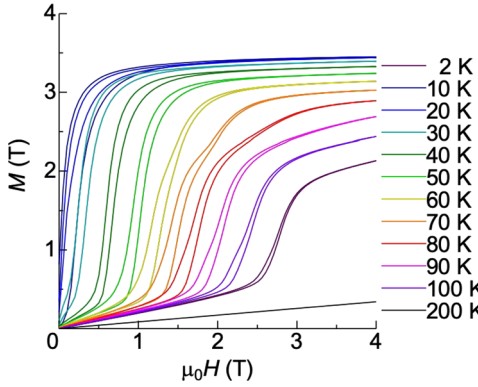

**Fig. 2 Magnetization curves in holmium single crystal.** The magnetic field was applied along the hexagonal [10$\bar{1}$0] direction at temperature from 2 to 200 K.

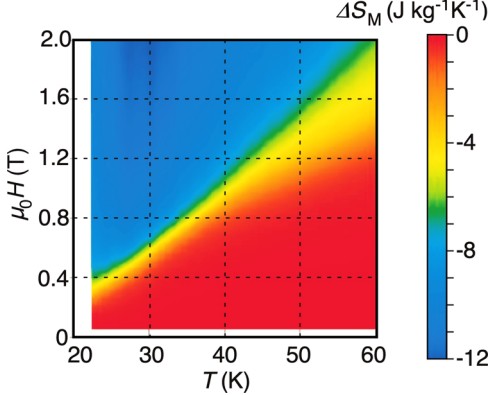

**Fig. 3 Magnetic entropy change $\Delta S_M$ as a function of temperature and magnetic field along the hexagonal [10$\bar{1}$0] direction in holmium single crystal.** The $\Delta S_M$ data are estimated by Eq. (1) with an integral range from $\mu_0 H_1 = 0$ T to $\mu_0 H_2 = \mu_0 H$ from the observed magnetization data.

hexagonal [10$\bar{1}$0] direction (Fig. 2). The changes are caused by the metamagnetic phase transition corresponding to a magnetic structure change from the spiral structure (spin-slip structure) to the FM ($T < 10$ K) or the other spiral structure ($T > 10$ K)[17,18]. The field-induced spiral structure, called the helifan structure, which has many propagation vectors including the higher harmonics, possesses a uniform magnetization with $M = 3$ T at $T = 10$ K, which is described in detail elsewhere[19,20]. It is important to note that the steep magnetization change occurs with a tiny magnetic field change at each temperature. The spin arrangement is therefore drastically changed only in the narrow field region.

From the magnetization at various temperatures shown in Fig. 2, we estimated the magnetic entropy change, $\Delta S_M(\mu_0 H_1, \mu_0 H_2)$, induced by changing the magnetic field from $\mu_0 H_1$ to $\mu_0 H_2$. Figure 3 shows $\Delta S_M(0, \mu_0 H)$ for the case when the magnetic field increases from zero. It can be seen that $\Delta S_M(0, \mu_0 H)$ remains almost zero below the metamagnetic transition field $\mu_0 H_c$; then, it abruptly decreases at $\mu_0 H_c$ and again becomes constant. The magnitude of $\Delta S_M(0, \mu_0 H = 2$ T) is roughly $-10$ J kg$^{-1}$ K$^{-1}$ at $\mu_0 H = 2$ T. The relative cooling parameter was estimated to ~0.5 kJ/kg at $\mu_0 H = 2$ T (see also in Supplementary Fig. 3). This magnitude is comparable to $\Delta S_M(0, \mu_0 H = 2$ T) for the typical magnetocaloric material HoAl$_2$ and is much smaller than the $\Delta S_M(0, \mu_0 H = 9$ T) of $-28$ J kg$^{-1}$ K$^{-1}$ for that same material. It is, however, too hasty to conclude that Ho is inferior to HoAl$_2$,

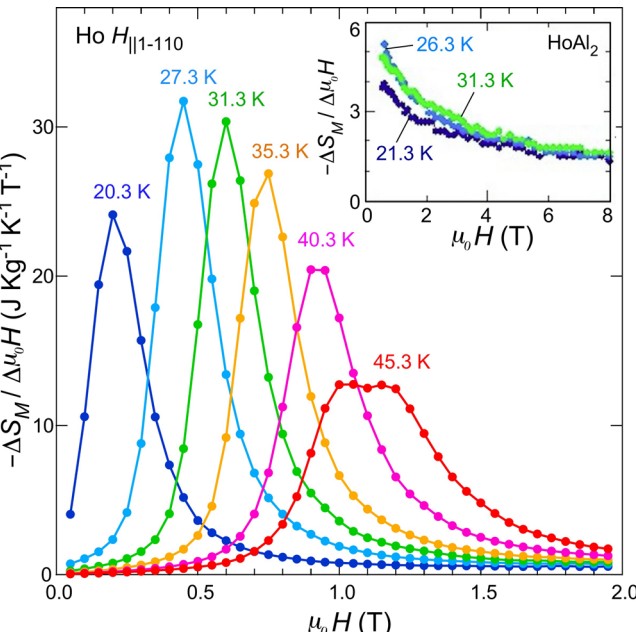

**Fig. 4 Magnetic field dependence of magnetic caloric efficiency, $-\Delta S_M/\Delta \mu_0 H$ (J kg$^{-1}$ K$^{-1}$), for the temperature range from 20 to 45 K in Ho.** The magnetic field was applied along the hexagonal [10$\bar{1}$0] direction. The $-\Delta S_M/\Delta \mu_0 H$ data are calculated by $-\Delta S_M$ divided by $\Delta \mu_0 H = 0.1$ T. The $-\Delta S_M$ is estimated by using Eq. (1) with the integration range from $\mu_0 H_1 = \mu_0 H$ to $\mu_0 H_2 = \mu_0 H + \Delta \mu_0 H$. For comparison between the present results and those for a typical magnetocaloric material, the efficiency for ferromagnetic HoAl$_2$ is also shown in the inset. The data for HoAl$_2$ were taken from a previous paper[31].

because the increase of $\Delta S_M(0, \mu_0 H)$ with magnetic field is gradual despite the fact that the energy cost of generating a magnetic field of 9 T is $(9/2)^2$ times larger than that for 2 T.

In this context, it is valuable to introduce a new figure of merit: the magnetocaloric efficiency index, defined as $-\Delta S_M(\mu_0 H_1, \mu_0 H_1 + \Delta \mu_0 H)/\Delta \mu_0 H$. As shown in Fig. 4, $-\Delta S_M(\mu_0 H_1, \mu_0 H_1 + \Delta \mu_0 H)/\Delta \mu_0 H$ of Ho is quite high only in the narrow range around $\mu_0 H_c(T)$. The height, $|\Delta S_M|/\Delta \mu_0 H = 32$ J kg$^{-1}$ K$^{-1}$ T$^{-1}$, is one order of magnitude larger than that of HoAl$_2$, as shown in the inset. This shows that MR can be made highly efficient not only by changing the magnetic field from zero but also by using the narrow range where the metamagnetic transition occurs. Needless to say, it is not practical if the absolute amplitude of the temperature change in a refrigeration cycle is too small in comparison with the temperature difference required for refrigeration devices, regardless of efficiency; hence, the actual temperature change caused by such magnetic field changes in Ho was examined next.

**Direct measurement of temperature change.** In order to confirm that an Ho sample is actually cooled/heated by the proposed process, we measured temperature change, $\Delta T$, through a temperature sensor placed directly on the Ho sample (within the adiabatic condition). The measurement was performed after applying a small magnetic field of $\mu_0 H_{ac} = 0.4$ T on a bias magnetic field of $\mu_0 H_0$. The $\mu_0 H_0$ was changed from 0 to 1.6 T every 0.1 T. In the case of $\mu_0 H_0 = 0.5$ T, the sample temperature rose from 28.2 to 29.7 K when the magnetic field was changed from 0.5 to 0.9 T, and returning the field to 0.5 T returned the temperature to 28.2 K (see the inset of Fig. 5b). Such thermal cycles with amplitudes approximately 1–1.5 K were observed for the conditions $\mu_0 H_0 \leq \mu_0 H_c \leq \mu_0 H_0 + \Delta \mu_0 H$, whereas small

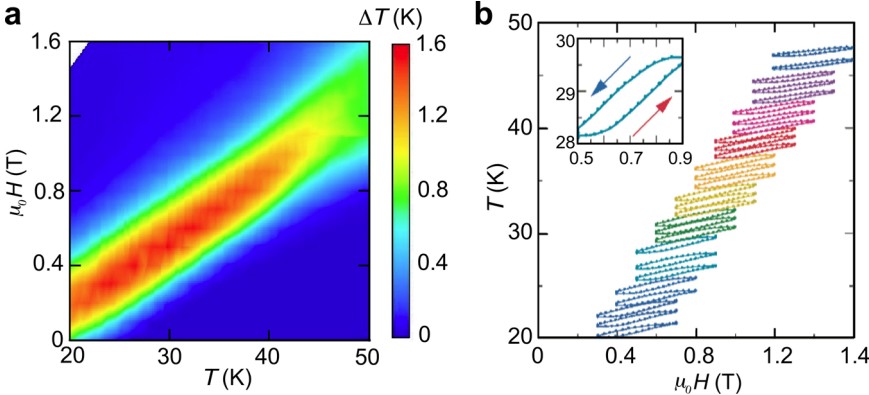

**Fig. 5 Results of the direct measurement of temperature change in holmium. a** Mapping of temperature change $\Delta T$ as a function of the bias magnetic field $\mu_0 H_0$ and temperature. **b** Typical heat cycles when the magnetic field rises from each $\mu_0 H_0$ by $\Delta \mu_0 H = 0.4$ T. The inset shows a magnification of the most efficient cycle for $\mu_0 H_0 = 0.5$ T and $T = 28.2$ K.

temperature changes occurred under other conditions, as shown in Fig. 5b.

As mapped in Fig. 5a, the bias field conditions were in agreement with predictions from $\Delta S_M(\mu_0 H_0, \mu_0 H_0 + \Delta \mu_0 H)$ (Fig. 3). In other words, adjusting the bias field enables us to cool the sample by 1–1.5 K at any temperature between 20 and 50 K. The relative amplitudes of these temperature changes correspond to several percent of the initial temperature and are comparable to those of conventional MR materials that have been examined for commercial refrigerators[21,22]. On the other hand, their absolute amplitudes seem much smaller in comparison with the requirements to cool a gas at 77 K to the hydrogen liquefaction temperature range. To solve this problem, we must consider utilization methods that can apply the excellent performance observed in Ho in practical MR systems.

**Practical system**. In this section, we discuss a practical MR system that uses small magnetic field changes and Ho metal. The MR system called active magnetic regenerator (AMR) was introduced by Barclay and Steyert in 1982 (ref. [23]). AMR refrigerator prototypes have been demonstrated using superconducting magnets (4–7 T)[24]. For the AMR system, it is necessary to distribute different types of magnetic materials that have different transition temperatures in order to obtain a large MCE in different temperature ranges (Fig. 6a)[23,25].

We propose an AMR system with Ho as shown in Fig. 6b. Because the temperature at which $\Delta T$ reaches its maximum value significantly depends on the bias magnetic field $\mu_0 H_0$ in Ho (Fig. 5), the system can be realized by a combination of a fixed $\mu_0 H_0$ distributed in space and an oscillating field $\Delta \mu_0 H$ distributed in time. The MR system is composed of (i) some pairs of permanent magnets with different constant magnetic fields, (ii) magnetocaloric material (Ho) movable in its position inside the system, and (iii) refrigerant (helium gas). Helium is a gas even below 20 K and chemically inactive. Aligning pairs of permanent magnets, one can realize spatial distribution of the bias magnetic field $\mu_0 H_0$ from 0.2 to 1.2 T. To change the magnetic field, the whole magnetic material is moved horizontally (in the figure) to the nearest permanent magnet position so that $\Delta \mu_0 H = \pm 0.2$ T. The magnetocaloric material is an assembly of small pieces of single crystals so that the gas refrigerant can flow among the pieces.

The operation of the system is performed as follows (Fig. 6c):

1. Moving the magnetocaloric material from the $i$th position to the neighboring permanent magnet adiabatically, one

obtains additional magnetic field at the $i$th position, $\mu_0 H_0^i + \Delta \mu_0 H$. Then, the temperature of the magnetocaloric material rises from the original value $T_0^i$ to $T_0^i + \Delta T_1^i$ due to the entropy change at position $i$. The additional quantity of heat expressed as $\Delta q_1^i$ is generated at the $i$th position.

2. The system is connected to one thermal bath (high-temperature side) and loses a total quantity of heat $\Delta Q_1 = \sum \Delta q_1^i$ through the gas refrigerant flow. The temperature of the magnetocaloric material is returned to $T_0^i$ at the $i$th position.

3. After returning to the adiabatic condition, the magnetocaloric material is positioned back to the origin, resulting in magnetic field change at the $i$th position back to $\mu_0 H_0^i$. Then, the temperature of the magnetocaloric material decreases from $T_0^i$ to $T_0^i - \Delta T_2^i$. The additional quantity of heat at the $i$th position is $-\Delta q_2^i$.

4. The system is connected to the other thermal bath (low-temperature side) and loses heat $\Delta Q_2 = \sum -\Delta q_2^i$, making the system temperature return to $T_0^i$. Then, the system cools the thermal bath by $\Delta Q_2 = \sum -\Delta q_2^i$.

In the present AMR system with Ho, it is not necessary to use different materials to obtain a large MCE at different positions. The temperature that shows a large $\Delta T$ can be varied by selecting the position of the material under different bias magnetic fields. The highest efficiency can be achieved in the wide temperature range from 20 to 50 K. In other words, the temperature dependence of $\Delta S_M$ can be effectively flat and the value can be kept at a high level over the entire the temperature range. Actually, the level for $\Delta \mu_0 H = 0.4$ T in Ho is comparable to $\Delta S_M$ that is achieved by using a well-designed complex material $(ErAl_2)_{0.312}/(HoAl_2)_{0.198}/(Ho_{0.5}Dy_{0.5}Al_2)_{0.490}$ for $\Delta \mu_0 H$ of 3 T[26]. Furthermore, the temperature gradient can be actively adjusted by controlling the field magnitude in accordance with continuous temperature readings.

**Potential materials**. We also investigated what types of materials, other than Ho, can be used for the presented AMR system. The material should show steep magnetization change with varying temperature and magnetic field. $ErCo_2$ is a known magnetocaloric material with a first-order magnetic phase transition, which has a large $-\Delta S_M = 33$ J kg$^{-1}$ K$^{-1}$ at 5 T[10]. Above the magnetic phase transition temperature of 32 K, $ErCo_2$ shows a metamagnetic phase transition; for example, $\mu_0 H_c = 1.6$ T at $T = 35$ K. The estimated maximum magnetocaloric efficiency index for $ErCo_2$ is $-\Delta S_M/\Delta \mu_0 H = 29$ J kg$^{-1}$ K$^{-1}$ T$^{-1}$ at $\mu_0 H = 2$ T and $T = 36$ K, which is comparable to the case of Ho. However, the field needed

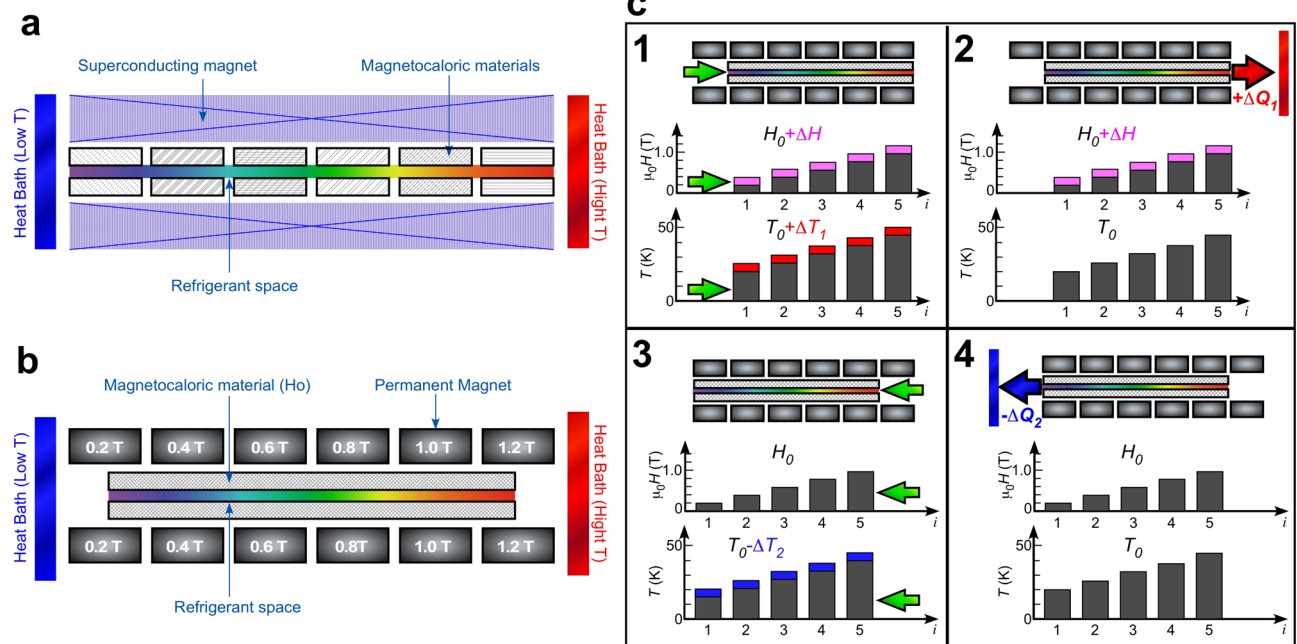

**Fig. 6 Schematic illustration of active magnetic regenerator (AMR) using permanent magnets and holmium. a** A cross-section of AMR setup[2,24]. The AMR constitutes a superconducting magnet, magnetocaloric materials, refrigerant space, and heat baths (low- and high-temperature sides). The magnetocaloric materials have different phase transition temperatures to obtain a large magnetocaloric effect at each position. **b** Possible setup of the proposed magnetic refrigeration (MR) system using a small change in magnetic field. The system constitutes pairs of permanent magnets, magnetocaloric material (holmium), refrigerant space, and heat baths. The set of permanent magnets generate a magnetic field gradient from 0.2 to 1.2 T as a bias magnetic field $\mu_0 H_0$ to change the magnetic phase transition temperature of the magnetocaloric material Ho. The magnetic field change $\Delta\mu_0 H$ is realized by mechanically moving the magnetocaloric material in the horizontal direction in this figure. **c** The MR cycle of the proposed AMR system. The gray color in the magnetic field and temperature graphs denotes the bias field $\mu_0 H_0$ and the original temperature $T_0$, respectively. The colored areas in the graphs are the changes in the magnetic field $\Delta\mu_0 H$ and temperature $\Delta T$.

to achieve metamagnetism in $ErCo_2$ above $T = 36$ K, such as $\mu_0 H_c = 3$ T at $T = 38.5$ K and $\mu_0 H_c = 4.2$ T at $T = 42.5$ K, might be too high to reach for permanent magnets.

Another rare-earth single metal, dysprosium (Dy), also shows metamagnetism with low phase transition fields ($\mu_0 H_c < 1.1$ T) for the temperature range between 85 and 178.5 K[27]. The magnetization curve with increasing field in Dy is similar to that of Ho, apart from the temperature range. We therefore expect that Dy has a comparable $-\Delta S_M / \Delta\mu_0 H$ value and would be useful for the presented AMR system for cooling in a different temperature range. There are some other materials that show metamagnetism. In $Co(S_{0.88}Se_{0.12})_2$, the working temperature range is from 10 to 60 K, and the field required ranges from 2 to 7 T[28,29]. $(Sm_{0.5}Gd_{0.5})_{0.55}Sr_{0.45}MnO_3$ also shows a steep magnetization change for the temperature range from 70 to 120 K, and the critical field is from 1 to 6 T for this range[11]. Therefore, these materials are suitable for neither the proposed system nor for hydrogen liquefaction due to the high working temperature and the high magnetic fields required. We thus argue that Ho metal is one of the best materials to use for the proposed AMR system with the combination of a normal electromagnet and permanent magnets.

In summary, MR research has long concentrated on searching for materials that show a large entropy change in a strong magnetic field. Here, we have proposed a system in which a tiny magnetic field change $\Delta\mu_0 H \leq 0.4$ T on top of a small magnetic field $\mu_0 H_0 < 1$ T achieves efficient MR by using a material (holmium) that exhibits steep magnetization changes in a small magnetic field. It should be emphasized that the present work does not find a large conventional MCE corresponding to $\Delta S_M$ in

a high magnetic field. What we have presented in this paper is that a small change in magnetic field can achieve highly efficient MR by using materials with steep magnetization changes such as holmium. In addition, in the present system it is not necessary to use a superconducting magnet with large operational costs. The proposed technique is a suitable alternative to conventional gas compression–based cooling for hydrogen liquefaction and could become a standard technique for low-temperature MR. Finally, it is hoped that this work opens a new research field focused on low-field MR for realizing low-cost MR in the near future.

## Method

A single crystal of Ho was purchased from the Crystal Base Company, Japan. The sample was cut into small pieces of 8 mg for magnetization measurement, and into plate-like shapes with dimensions of approximately $5 \times 5 \times 0.5$ mm$^3$ and mass of 276 mg for the thermal measurement. For measuring magnetization, we used the magnetic properties measurement system manufactured by Quantum Design (QD). For direct measurement of temperature change, a zirconium oxy-nitride thin-film thermometer called Cernox$^{TM}$ (CX-SD, Lake Shore Cryotronics) was placed on the large surface of a plate-shaped sample with a small amount of Apiezon N grease. The sample assembly was inserted into QD physical properties measurement system (PPMS). Temperature and magnetic field were controlled by the PPMS. The sample space was continuously pumped by a turbo pump, and the pressure was kept below $10^{-5}$ Pa ($10^{-7}$ torr) to reach adiabatic conditions.

## Data availability

The data that support the plots within this paper and other findings of this study are available from the corresponding author upon reasonable request.

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

## Acknowledgements

This work was supported by JSPS KAKENHI (Grant No. 17KK0099 and 19H04400) and JST-Mirai Program, Japan (Grant No. JPMJMI18A3).

## Author contributions

N.T. and H.M. carried out the magnetization measurements and the thermal measurements, analyzed the data, and wrote the manuscript.

## Competing interests

The authors declare no competing interests.
