## [Peer Review File · Nature Communications]

Reviewers' Comments:

Reviewer #1:

None

Reviewer #2:

Remarks to the Author:

The authors present an experimental work involving a magnetic refrigeration using holmium at low temperature. Precisely, they propose a cooling technique in which small magnetic field changes, $\Delta\mu_0H \leq 0.4$ T, can obtain a cooling efficiency of $-\Delta SM/\Delta\mu_0H = 32$ J kg⁻¹K⁻¹T⁻¹, which is one order of magnitude higher than has been achieved using typical magnetocaloric materials. In addition, they claim that this method leads to a high cooling efficiency. The manuscript is presented clearly. It includes interesting results on cooling refrigeration that can be useful for scientific community working on cryogenic refrigeration. Satisfactory explanations are given as well. The scientific quality of this article is high enough for the publication in the Journal. However, some minor revisions and clarifications are needed:

1. For a clear idea on material magnetism, both magnetizations $M(T)$ under magnetic field of 0,05T and $M(H)$ at lower temperature (< 10 K) must be given
2. As tradition, to evaluate the magnetocaloric effect of Ho metal for comparison, the Relative Cooling Power has to be given for their highest magnetic field (1.2T).
3. The authors seem to claim that using Ho metal giving a high magnetocaloric effect is a discovery. I do not agree with their claim, since it is known that Ho is the atom in periodic table of elements showing the highest magnetic moment (10 μ_B). Therefore, a high magnetocaloric effect is expected. The authors have to reorganize their text in order to present that using Ho metal is only a good choice.
4. The authors report that obtaining a high magnetocaloric effect leads to a High-Efficiency Magnetic Refrigeration. I do not agree with this idea, since a High-Efficiency Magnetic Refrigeration needs a high working refrigeration temperature which is the adiabatic temperature in a single material case or a span temperature T_{span} in composite material case. This working refrigeration T is proportional to the ratio $\Delta SM/CB$ (CB is the specific heat capacity under a constant magnetic field). If CB value is high, the working refrigeration temperature will be low even if ΔSM is high. The authors have to present an adequate comment in the introduction related to this issue.
5. Unlike magnetic ceramics such manganites, the rare earth metals show a strong corrosion during the use. How authors can resolve this problem? In addition, due to their low specific heat capacity, the manganites can give similar working refrigeration temperature at any low temperature. Adequate comment related to this issue must be added in the introduction to justify their Ho choice.
6. The authors have to explain why the unity of the entropy is not specified in Fig. 1 ?
7. In Fig 2, the unity of $M(T)$ has to be specified in uem/g.

Reviewer #3:

Remarks to the Author:

Authors reported a study on High-Efficiency Magnetic Refrigeration Using Holmium. The proposed technique and the related experimental results are interesting and have been explained. If this can be applicable in practice, it should attract lots of attention in this field. So, it therefore deserves publication. However, the references listed are very old to me (the latest one was published in 2015). The authors need add more discussions on their results related to recent progress on this field. Once this is done, the impact of the work will be visible. So, I recommend 'Major Revision' for publication.

Dear Nature Communications editors,

Thank you very much for forwarding the referees' comments on our manuscript. We are pleased to receive the two referee's positive comments for publication in the journal. The point-by-point responses to the referees are shown below. The red colored text represents corrected points in the revised manuscript. The manuscript with corrections colored in red is attached after the responses.

Yours sincerely,
Noriki Terada and Hiroaki Mamiya

Reviewer #2

Comment 0)

The authors present an experimental work involving a magnetic refrigeration using holmium at low temperature. Precisely, they propose a cooling technique in which small magnetic field changes, $\Delta\mu_0H \leq 0.4$ T, can obtain a cooling efficiency of $-\Delta S_M/\Delta\mu_0H = 32$ J kg⁻¹K⁻¹T⁻¹, which is one order of magnitude higher than has been achieved using typical magnetocaloric materials. In addition, they claim that this method leads to a high cooling efficiency. The manuscript is presented clearly. It includes interesting results on cooling refrigeration that can be useful for scientific community working on cryogenic refrigeration. Satisfactory explanations are given as well. The scientific quality of this article is high enough for the publication in the Journal.

Response 0)

Firstly, we acknowledge that the referee has read and reviewed our manuscript carefully and precisely. We also strongly appreciate that the referee agrees with the importance of our work.

Comment 1)

However, some minor revisions and clarifications are needed:

1. For a clear idea on material magnetism, both magnetizations $M(T)$ under magnetic field of 0,05T and $M(H)$ at lower temperature ($< 10K$) must be given

Response 1)

Thank you very much for the constructive comment to improve the manuscript. $M(T)$ under low field has been reported in Ref. [13], and $M(H)$ below 10 K (2 K) is also shown in Fig. 2 in the manuscript. **In order to make the material magnetism of Ho clearer, we add some explanation for the phase transition in Ho in the 2nd paragraph of page 3. We also added the $M(T)$ in 0.01 T as supplementary Figure 1 in the supplementary information.**

Comment 2)

2.As tradition, to evaluate the magnetocaloric effect of Ho metal for comparison, the Relative Cooling Power has to be given for their highest magnetic field (1.2T).

Response 2)

We added the Relative Cooling Power value for 2 T in the 2nd paragraph of page 4, and added ΔS_M data as a function of temperature at 2 T in the supplementary Figure 3 in the supplementary information.

Comment 3)

3. The authors seem to claim that using Ho metal giving a high magnetocaloric effect is a discovery. I do not agree with their claim, since it is known that Ho is the atom in periodic table of elements showing the highest magnetic moment (10 μ_B). Therefore, a high magnetocaloric effect is expected. The authors have to reorganize their text in order to present that using Ho metal is only a good choice.

Response 3)

We agree with the referee's suggestion. Actually, our work did not find that Ho metal has a large conventional magnetocaloric effect (corresponding to ΔS_M value), which has been already reported in Ref[17]. Our argument in this paper is that a small change of magnetic field can achieve a high efficient magnetic refrigeration by using materials with steep magnetization changes such as Ho, as suggested by the referee comment. **In order to avoid any confusions, we added a statement in the conclusion section.**

Comment 4)

4. The authors report that obtaining a high magnetocaloric effect leads to a High-Efficiency Magnetic Refrigeration. I do not agree with this idea, since a High-Efficiency Magnetic Refrigeration needs a high working refrigeration temperature which is the adiabatic temperature in a single material case or a span temperature T_{span} in composite material case. This working refrigeration T is proportional to the ratio $\Delta SM/CB$ (CB is the specific heat capacity under a constant magnetic field). If CB value is high, the working refrigeration temperature will be low even if ΔSM is high. The authors have to present an adequate comment in the introduction related to this issue.

Response 4)

We appreciate the referee's suggestion that the adiabatic temperature change ΔT is an important quantity to evaluate a magnetic refrigeration. In this reason, we directly measured ΔT as well as ΔS_M (derived from the magnetization data). **Along the referee's suggestion, we added some statements regarding the adiabatic temperature proportional to $\Delta SM/CB$ in the introduction section (page 1).**

Comment 5)

5. Unlike magnetic ceramics such manganites, the rare earth metals show a strong corrosion during the use. How authors can resolve this problem? In addition, due to their low specific heat capacity, the manganites can give similar working refrigeration temperature at any low temperature. Adequate comment related to this issue must be added in the introduction to justify their Ho choice.

Response 5)

We appreciate the referee's comment on a problem for a corrosion of the rare earth metal. As suggested by the referee, manganites such as $(\text{Sm}_{0.5}\text{Gd}_{0.5})_{0.55}\text{SrMnO}_3$ (Ref. [12]) are also candidates materials to use for MR with metamagnetism proposed in the paper. However, as discussed in the discussion (the first paragraph in page 8), the working temperature range (70 K ~ 20K) is much higher than the hydrogen liquefaction temperature 20 K, and a high magnetic field (1 T ~ 6 T) is also needed for $(\text{Sm}_{0.5}\text{Gd}_{0.5})_{0.55}\text{SrMnO}_3$. **We added some statements related to the reason why we chose holmium for this study in the 3rd paragraph in page 2.**

As for the corrosion, in the present case, we basically consider helium gas as a refrigerant, which is chemically inactive. **We added a statement related to this in the second paragraph of page 6.**

Comment 6)

6. The authors have to explain why the unity of the entropy is not specified in Fig. 1 ?

Response 6)

Fig. 1(a) shows the calculation data taken from theoretical simulation for ferromagnetic case in Ref.[10]. As described as "Schematic illustration of MR cycle" in the caption of Fig. 1(b), it shows a schematic drawing (without units) for a case of metamagnetic antiferromagnet to roughly explain the idea of this work.

Comment 7)

7. In Fig 2, the unity of $M(T)$ has to be specified in uem/g.

Response 7)

We added the $M(T)$ data in emu/g unit in supplementary Figure 2 in the supplementary information.

Reviewer #3

Comment 0)

Authors reported a study on High-Efficiency Magnetic Refrigeration Using Holmium. The proposed technique and the related experimental results are interesting and have been explained. If this can be applicable in practice, it should attract lots of attention in this field. So, it therefore deserves publication.

Response 0)

We acknowledge that the referee has read and reviewed our manuscript carefully and precisely. We also strongly appreciate that the referee agrees with the importance of our work.

Comment 1)

However, the references listed are very old to me (the latest one was published in 2015). The authors need add more discussions on their results related to recent progress on this field. Once this is done, the impact of the work will be visible. So, I recommend 'Major Revision' for publication.

Response 1)

Thank you very much for the constructive comment. Along the referee's suggestion, we added a paragraph in the introduction section (4th paragraph in page 2), to explain two recent papers about magnetocaloric effect of HoB₂ (Ref.[8]) and RB₄ (Ref. [9]) published in 2020.

High-Efficiency Magnetic Refrigeration Using Holmium

Noriki Terada* and Hiroaki Mamiya**

¹*National Institute for Materials Science,
Sengen 1-2-1, Tsukuba, Ibaraki 305-0047, Japan*

(Dated: September 24, 2020)

Abstract

Magnetic refrigeration (MR), which is a method to cool matter using a magnetic field, has been studied for application to refrigeration near room temperature. Recently, MR research has also focused on a target temperature of 20 K for hydrogen liquefaction. Most research to date has employed high magnetic fields (at least 5 T) to obtain a large entropy change, which requires large energy costs to operate a superconducting magnet. Alternatively, we propose here a highly efficient cooling technique in which small magnetic field changes, $\Delta\mu_0 H \leq 0.4$ T, can obtain a cooling efficiency of $-\Delta S_M/\Delta\mu_0 H = 32$ J kg⁻¹K⁻¹T⁻¹, which is one order of magnitude higher than has been achieved using typical magnetocaloric materials. Our method uses holmium, which exhibits a steep magnetization change with varying temperature and magnetic field. The proposed technique can be implemented with permanent magnets, making it a suitable alternative to conventional gas compression-based cooling for hydrogen liquefaction.

Introduction

Storing energy is one of the most important issues for using renewable energy resources. Although liquid hydrogen is expected to be used for energy storage, at present, a gas (helium or hydrogen) compression cycle cooling technique is widely used for liquefaction of hydrogen, which has high operation costs.[1]

Magnetic refrigeration (MR) is an alternative refrigeration technique to gas compression.[2] MR, which is a method to cool matter using a magnetic field, was discovered more than 100 years ago.[3, 4] For the past 50 years, MR has been intensively studied for application to refrigeration near room temperature.[5] More recently, the MR research field has expanded to investigations of low target temperature for the purpose of hydrogen liquefaction, which requires refrigeration to the condensation temperature of 20 K.[2] MR cools matter by using magnetic entropy changes that occur when a magnetic field is applied, a phenomenon called the magnetocaloric effect (MCE).[6] The MCE, also known as adiabatic diamagnetization among low temperature physicists,[7] is quantified by the thermodynamic formula based on the Maxwell relation.

$$\Delta S_M = \mu_0 \int_{\mu_0 H_1}^{\mu_0 H_2} \left(\frac{\partial M(T, H)}{\partial T} \right)_H dH \quad (1)$$

where ΔS_M is the magnetic entropy change during magnetic field (H) change from H_1 to H_2 and M is the magnetization. **Adiabatic temperature change defined by $\Delta T \equiv \Delta S_M/C$ (C is a specific heat) is also an important quantity to evaluate MR efficiency of a material.**

There are some candidate materials in the MR for the purpose of the hydrogen liquefaction. For example, HoB_2 has been recently reported to show the largest MCE $\Delta S_M = 40.1 \text{ J kg}^{-1}\text{K}^{-1}$ in the MR materials with the phase transition temperature near the hydrogen liquefaction.[8] RB_4 ($R = \text{Dy, Ho}$) have also been recently discovered to show a large magnetic entropy change achieved by a strong coupling between spin and quadrupolar degrees of freedom.[9] However, to date, high magnetic fields, not lower than 2 T, have been necessary to obtain a large entropy change, requiring large energy costs to operate a superconducting magnet.

A large ΔS_M is related to a large magnetization change with varying temperature. For this reason, ferromagnetic (FM) materials at their Curie temperature T_C have conventionally been considered for MR applications. Figure 1(a) shows a simple MR cycle $A \rightarrow B \rightarrow C \rightarrow A$

of a model FM material. Most reduction of the magnetic entropy can be realized only by application of a magnetic field for which the Zeeman energy becomes comparable to the thermal energy at T_C . The required magnetic field is typically more than several Teslas in the temperature range between the hydrogen (20.3 K) and nitrogen (77 K) liquefaction temperatures. Thus, use of stronger magnetic fields leading to larger ΔS_M in FM materials has been considered more suitable for hydrogen liquefaction, even though the energy cost of generating the magnetic field steeply increases in proportion to the square of the field amplitude.

In this study, we note that magnetic materials exhibiting a large magnetization jump (meta-magnetization) show a large magnetic entropy change ΔS_M in a narrow magnetic field range. In this case, the magnetization change achieved in minor cycles, $A' \rightarrow B' \rightarrow C' \rightarrow A'$, using a tiny field change can be almost identical to those obtained in the major $A \rightarrow B \rightarrow C \rightarrow A$ cycle (shown in Fig. 1(b)) and cause a sufficient MR effect. **There are some examples to show a large magnetization change in a narrow magnetic field range; for example, rare earth intermetallic ErCo_2 [11] and oxide ceramic $(\text{Sm}_{0.5}\text{Gd}_{0.5})_{0.55}\text{Sr}_{0.45}\text{MnO}_3$. [12] In both cases, however, it is necessary to apply a magnetic field higher than 3 T to change the magnetization. On the other hand, the rare single metal holmium (Ho) exhibits a large magnetization change in a smaller magnetic field lower than 1 T for the temperature range from 20 K to 50 K as described below. Therefore, we selected the holmium (Ho) and studied its magnetocaloric properties.** Consequently, the large MCE can be driven by a tiny vibration of the magnetic field, resulting in MR without a large energy cost.

Ho is known as an antiferromagnetic (AFM) material with helicoidal spin structure.[13–16] **The phase transition from the paramagnetic to AFM helicoidal phases occurs at $T = 132$ K, which is followed by appearance of the ferromagnetic phase below $T = 20$ K in zero magnetic field.**[13] The MCE of Ho in a strong magnetic field (6 T) has been reported.[17] However, in some AFM cases, such as Ho, a large and steep magnetization change, caused by the metamagnetic transition, occurs even when a small magnetic field is applied. The critical field, $\mu_0 H_c$, that results in a large magnetization change is much smaller than that generated by a superconducting magnet; for example, $\mu_0 H_c = 0.2$ T at $T = 20$ K, and $\mu_0 H_c = 1.0$ T at $T = 50$ K. A large change can also be driven by a tiny variation of magnetic field $\Delta\mu_0 H \leq \sim 0.4$ T at each $\mu_0 H_c$. We thus anticipate that an efficient MCE can be obtained in Ho without using a strong magnetic field.

Results and Discussion

Magnetic entropy change

Steep and large magnetization changes were observed in Ho single crystals when a magnetic field was applied along the hexagonal $[10\bar{1}0]$ direction (Fig. 2). The changes are caused by the metamagnetic phase transition corresponding to a magnetic structure change from the spiral structure (spin-slip structure) to the FM ($T < 10$ K) or the other spiral structure ($T > 10$ K).[18, 19] The field-induced spiral structure, called the helifan structure, which has many propagation vectors including the higher harmonics, possesses a uniform magnetization with $M = 3$ T at $T = 10$ K, which is described in detail elsewhere.[20, 21] It is important to note that the steep magnetization change occurs with a tiny magnetic field change at each temperature. The spin arrangement is therefore drastically changed only in the narrow field region.

From the magnetization at various temperatures shown in Fig. 2, we estimated the magnetic entropy change, $\Delta S_M(\mu_0 H_1, \mu_0 H_2)$, induced by changing the magnetic field from $\mu_0 H_1$ to $\mu_0 H_2$. Figure 3 shows $\Delta S_M(0, \mu_0 H)$ in the case that the magnetic field increases from zero. It can be seen that $\Delta S_M(0, \mu_0 H)$ remains almost zero below the metamagnetic transition field $\mu_0 H_c$; then, it abruptly decreases at $\mu_0 H_c$ and again becomes constant. The magnitude of $\Delta S_M(0, \mu_0 H = 2$ T) is roughly -10 J kg $^{-1}$ K $^{-1}$ at $\mu_0 H = 2$ T. **The Relative Cooling Parameter was estimated to ~ 0.5 kJ/kg at $\mu_0 H = 2$ T. (see also in supplementary Figure 3)** This magnitude is comparable to $\Delta S_M(0, \mu_0 H = 2$ T) for the typical magnetocaloric material HoAl $_2$ and is much smaller than the $\Delta S_M(0, \mu_0 H = 9$ T) of -28 J kg $^{-1}$ K $^{-1}$ for that same material. It is, however, too hasty to conclude that Ho is inferior to HoAl $_2$, because the increase of $\Delta S_M(0, \mu_0 H)$ with magnetic field is gradual despite the fact that the energy cost of generating a magnetic field of 9 T is $(9/2)^2$ times larger than that for 2 T.

In this context, it is valuable to introduce a new figure of merit: the magnetocaloric efficiency index, defined as $-\Delta S_M(\mu_0 H_1, \mu_0 H_1 + \Delta\mu_0 H)/\Delta\mu_0 H$. As shown in Fig. 4, $-\Delta S_M(\mu_0 H_1, \mu_0 H_1 + \Delta\mu_0 H)/\Delta\mu_0 H$ of Ho is quite high only in the narrow range around $\mu_0 H_c(T)$. The height, $|\Delta S_M|/\Delta\mu_0 H = 32$ J kg $^{-1}$ K $^{-1}$ T $^{-1}$, is one order of magnitude larger than that of HoAl $_2$, as shown in the inset. This shows that MR can be made highly efficient

not only by changing the magnetic field from zero but also by using the narrow range where the metamagnetic transition occurs. Needless to say, it is not practical if the absolute amplitude of the temperature change in a refrigeration cycle is too small in comparison with the temperature difference required for refrigeration devices, regardless of efficiency; hence, the actual temperature change caused by such magnetic field changes in Ho was examined next.

Direct measurement of temperature change

In order to confirm that a Ho sample is actually cooled/heated by the proposed process, we measured temperature change, ΔT , by reading a temperature sensor placed directly on the Ho sample (within the adiabatic condition). The measurement was performed by applying a small magnetic field of $\mu_0 H_{ac} = 0.4$ T on a bias magnetic field of $\mu_0 H_0$. The $\mu_0 H_0$ was changed from 0 to 1.6 T every 0.1 T. In the case of $\mu_0 H_0 = 0.5$ T, the sample temperature rose from 28.2 K to 29.7 K when the magnetic field was changed from 0.5 T to 0.9 T, and returning the field to 0.5 T returned the temperature to 28.2 K (see the inset of Fig. 5(b)). Such thermal cycles with amplitudes approximately 1 – 1.5 K were observed for the conditions $\mu_0 H_0 \leq \mu_0 H_c \leq \mu_0 H_0 + \Delta\mu_0 H$, whereas small temperature changes occurred under other conditions, as shown in Fig. 5(b).

As mapped in Fig. 5(a), the bias field conditions were in agreement with predictions from $\Delta S_M(\mu_0 H_0, \mu_0 H_0 + \Delta\mu_0 H)$ (Fig. 3). In other words, adjusting the bias field enables us to cool the sample by 1 – 1.5 K at any temperature in the range between 20 K and 50 K. The relative amplitudes of these temperature changes correspond to several percent of the initial temperature and are comparable to those of conventional MR materials that have been examined for commercial refrigerators.[24] On the other hand, their absolute amplitudes seem much smaller in comparison with the requirements to cool a gas at 77 K to the hydrogen liquefaction temperature range. To solve this problem, we must consider utilization methods that can apply the excellent performance observed in Ho in practical MR systems.

Practical system

In this section, we discuss a practical MR system that uses small magnetic field changes and Ho metal. The MR system called Active Magnetic Regenerator (AMR) was introduced by Barclay and Steyert in 1982.[25] AMR refrigerator prototypes have been demonstrated using superconducting magnets (4–7 T).[26] For the AMR system, it is necessary to distribute different types of magnetic materials that have different transition temperatures, in order to obtain a large MCE in different temperature ranges (Fig. 6(a)).[25, 27]

We propose the AMR system with Ho shown in Fig. 6(b). Because the temperature at which ΔT reaches its maximum value significantly depends on the bias magnetic field $\mu_0 H_0$ in Ho (Fig. 5), the system can be realized by a combination of a fixed $\mu_0 H_0$ distributed in space and an oscillating field $\Delta\mu_0 H$ distributed in time. The MR system is composed of (i) some pairs of permanent magnets with different constant magnetic fields, (ii) magnetocaloric material (Ho) movable in its position inside the system, and (iii) refrigerant (Helium gas). **Helium is a gas even below 20 K and chemically inactive.** Aligning pairs of permanent magnets, one can realize spatial distribution of the bias magnetic field $\mu_0 H_0$ from 0.2 T to 1.2 T. To change the magnetic field, the whole magnetic material is moved horizontally (in the figure) to the nearest permanent magnet position so that $\Delta\mu_0 H = \pm 0.2$ T. The magnetocaloric material is an assembly of small pieces of single crystals so that the gas refrigerant can flow among the pieces.

The operation of the system is performed as follows (Fig. 6(c)):

1. Moving the magnetocaloric material from the i -th position to the neighboring permanent magnet adiabatically, one obtains additional magnetic field at the i -th position, $\mu_0 H_0^i + \Delta\mu_0 H$. Then, the temperature of the magnetocaloric material rises from the original value T_0^i to $T_0^i + \Delta T_1^i$ due to the entropy change at position i . The additional quantity of heat expressed as Δq_1^i is generated at the i -th position.
2. The system is connected to one thermal bath (high-temperature side) and loses a total quantity of heat $\Delta Q_1 = \sum \Delta q_1^i$ by the gas refrigerant flow. The temperature of the magnetocaloric material is returned to T_0^i at the i -th position.
3. After returning to the adiabatic condition, the magnetocaloric material is positioned back to the origin, resulting in magnetic field change at the i -th position back to $\mu_0 H_0^i$. Then, the temperature of the magnetocaloric material decreases from T_0^i to $T_0^i - \Delta T_2^i$. The additional

quantity of heat at the i -th position is $-\Delta q_2^i$.

4. The system is connected to the other thermal bath (low-temperature side) and loses heat $\Delta Q_2 = \sum -\Delta q_2^i$, making the system temperature return to T_0^i . Then, the system cools the thermal bath by $\Delta Q_2 = \sum -\Delta q_2^i$.

In the present AMR system with Ho, it is not necessary to use different materials to obtain a large MCE at different positions. The temperature that shows a large ΔT can be varied by selecting the position of the material under different bias magnetic fields. The highest efficiency can be achieved in the wide temperature range from 20 K to 50 K. In other words, the temperature dependence of ΔS_M can be effectively flat and the value can be kept at a high level over the entire the temperature range. Actually, the level for $\Delta\mu_0 H = 0.4$ T in Ho is comparable to ΔS_M that is achieved by using a well-designed complex material $(\text{ErAl}_2)_{0.312}/(\text{HoAl}_2)_{0.198}/(\text{Ho}_{0.5}\text{Dy}_{0.5}\text{Al}_2)_{0.490}$ for $\Delta\mu_0 H$ of 3 T.[28] Furthermore, the temperature gradient can be actively adjusted by controlling the field magnitude in accordance with continuous temperature readings.

Potential materials

We also investigated what types of materials, other than Ho, can be used for the presented AMR system. The material should show steep magnetization change with varying temperature and magnetic field. ErCo_2 is a known magnetocaloric material with a first-order magnetic phase transition, which has a large $-\Delta S_M = 33$ J $\text{kg}^{-1}\text{K}^{-1}$ at 5 T.[11] Above the magnetic phase transition temperature 32 K, ErCo_2 shows a metamagnetic phase transition; for example, $\mu_0 H_c = 1.6$ T at $T = 35$ K. The estimated maximum magnetocaloric efficiency index for ErCo_2 is $-\Delta S_M/\Delta\mu_0 H = 29$ J $\text{kg}^{-1}\text{K}^{-1}\text{T}^{-1}$ at $\mu_0 H = 2$ T and $T = 36$ K, which is comparable to the case of Ho. However, the field needed to achieve metamagnetism in ErCo_2 above $T = 36$ K, such as $\mu_0 H_c = 3$ T at $T = 38.5$ K and $\mu_0 H_c = 4.2$ T at $T = 42.5$ K, might be too high to reach by using permanent magnets.

Another rare earth single metal, dysprosium (Dy), also shows metamagnetism with low phase transition fields ($\mu_0 H_c < 1.1$ T) for the temperature range between 85 K and 178.5 K.[29] The magnetization curve with increasing field in Dy is similar to that of Ho apart from the temperature range. We therefore expect that Dy has a comparable $-\Delta S_M/\Delta\mu_0 H$ value and would be useful for the presented AMR system for cooling in a different temperature

range. There are some other materials that show metamagnetism. In $\text{Co}(\text{S}_{0.88}\text{Se}_{0.12})_2$, the working temperature range is from 10 K to 60 K, and the field required ranges from 2 T to 7 T. [30, 31] $(\text{Sm}_{0.5}\text{Gd}_{0.5})_{0.55}\text{Sr}_{0.45}\text{MnO}_3$ also shows a steep magnetization change for the temperature range from 70 K to 120 K, and the critical field is from 1 T to 6 T for this range.[12] Therefore, these materials are not suitable for the presently proposed system or hydrogen liquefaction due to the high working temperature and the high required fields. We thus argue that Ho metal is one of the best materials to use for the presented AMR system with the combination of a normal electromagnet and permanent magnets.

Conclusion

In summary, MR research has long concentrated on searching for materials that show a large entropy change in a strong magnetic field. Here, we have proposed a system in which a tiny magnetic field change $\Delta\mu_0 H \leq 0.4$ T on top of a small magnetic field $\mu_0 H_0 < 1$ T achieves efficient MR by using a material (holmium) that exhibits steep magnetization changes in a small magnetic field. **It should be emphasized that the present work does not find a large conventional magnetocaloric effect, corresponding to ΔS_M in a high magnetic field. We have presented in this paper that a tiny change of magnetic field can achieve the high efficient MR by using materials with steep magnetization changes such as holmium.** In addition, it is not necessary to use a superconducting magnet with large operation cost in the present system. The proposed technique is a suitable alternative to conventional gas compression-based cooling for hydrogen liquefaction and could become a standard technique for low-temperature MR. Finally, it is hoped that this work opens a new research field focused on low-field MR for realizing low-cost MR in the near future.

Method

A single crystal of Ho was purchased from the Crystal Base company, Japan. The sample was cut into small pieces of 8 mg for magnetization measurement, and into plate-like shapes with dimensions approximately $5 \times 5 \times 0.5$ mm³ and 276 mg mass for the thermal measurement. For magnetization measurement, we used the magnetic properties measurement system (MPMS) manufactured by Quantum Design (QD). For direct measurement of tem-

perature change, a zirconium oxy-nitride thin-film thermometer called CernoxTM (CX-SD, Lake Shore Cryotronics) was placed on the large surface of a plate-shaped sample with a small amount of Apiezon N grease. The sample assembly was inserted into the QD physical properties measurement system (PPMS). Temperature and magnetic field were controlled by the PPMS. The sample space was continuously pumped by a turbo pump and the pressure was kept below 10^{-5} Pa (10^{-7} torr), to reach adiabatic conditions.

Data availability

The data that support the plots within this paper and other findings of this study are available from the corresponding author upon reasonable request.

Code availability

No computer code was used in this study.

-
- [1] Cipriani, G., Di Dio, V., Genduso, F., La Cascia, D., Liga, R., Miceli, R. & Galluzzo, G. R. Perspective on hydrogen energy carrier and its automotive applications. *Int. J. Hydrogen Energy* **39**, 8482–8494, (2014).
 - [2] Numazawa, T., Kamiya, K., Utaki, T. & Matsumoto, K. Magnetic refrigerator for hydrogen liquefaction. *Cryogenics* **62**, 185–192 (2014).
 - [3] Warburg, E. Magnetische Untersuchungen. Uebereinigende Wirkungen der Coercitivkraft. *Ann. Phys. (Leipzig)* **249**, 141–164 (1881).
 - [4] Weiss, P. & Piccard, A. Le phenomene magnetocalorique. *J. Phys. (Paris)*, 5th Ser. 7, 103–109 (1917).
 - [5] Gschneidner Jr., K. A. & Pecharsky, V. K. Thirty years of near room temperature magnetic cooling: Where we are today and future prospects. *Int. J. Refrigeration* **31**, 945–961 (2008).
 - [6] Franco, V., Blázquez, J. S., Ingale, B., & Conde, A. The magnetocaloric effect and magnetic refrigeration near room temperature: Materials and models. *Annu. Rev. Mater. Res.* **42**, 305–342 (2012).

- [7] Kurti, N., Robinson, F. N. H., Simon, F. & Spohr, D. A. Nuclear cooling. *Nature* **178**, 450–453 (1956).
- [8] Baptista de Castro, P., Terashima K., Yamamoto T. D., Hou Z., Iwasaki S., Matsumoto R., Adachi S., Saito Y., Song P., Takeya H. & Takano Y. *Nature Asia Mater.* **12**, 35 (2020).
- [9] Song, M. S., Cho, K. K., Kang B.Y., Lee S. B. & Cho B. K. *Sci. Rep.* **10**, 803 (2020).
- [10] Tamura, R., Ohno, T. & Kitazawa, H. A generalized magnetic refrigeration scheme. *Appl. Phys. Lett.* **104**, 052415 (2014).
- [11] Wada, H., Tanabe, Y., Shiga, M., Sugawara, H. & Sato, H. Magnetocaloric effects of Laves phase $\text{Er}(\text{Co}_{1-x}\text{Ni}_x)_2$ compounds. *J. Alloys Compounds* **316**, 245–249 (2001).
- [12] Tomioka, Y., Okimoto, Y., Jung, J. H., Kumai, R. & Tokura, Y. Critical control of competition between metallic ferromagnetism and charge/orbital correlation in single crystals of perovskite manganites. *Phys. Rev. B* **68**, 094417 (2003).
- [13] Rhodes, B. L., Legvold, S. & Spedding, F. H. Magnetic properties of holmium and thulium metals. *Phys. Rev.* **109**, 1547–1550 (1958).
- [14] Strandburg, D. L., Legvold, S. & Spedding, F. H. Electrical and magnetic properties of holmium single crystals. *Phys. Rev.* **127**, 2046–2051 (1962).
- [15] Jensen, J. & Mackintosh, A. R. Helifan: A new type of magnetic structure. *Phys. Rev. Lett.* **64**, 2699–2702 (1990).
- [16] Gibbs, D., Moncton, D. E., DAmico, K. L., Bohr, J. & Grier, B. H. Magnetic x-ray scattering studies of holmium using synchrotron radiation. *Phys. Rev. Lett.* **55**, 234–237 (1985).
- [17] Nikitin, S. A., Andreyenko, A. S., Tishin, A. M., Arkharov, A. M. & Zherdev, A. A. Magnetocaloric effect in heavy rare-earth metals. *Phys. Met. Metall.* **60**, 56–61 (1985).
- [18] Cowley, R. A., Jehan, D. A., McMorro, D.F. & McIntyre, G. J. Evidence for a devil’s staircase in holmium produced by an applied magnetic field. *Phys. Rev. Lett.* **66**, 1521–1524 (1990).
- [19] Jehan, D. A. , McMorro, D. F., Cowley, R. A. & McIntyre, G. J. The magnetic structure of holmium in an easy-axis magnetic field. *Europhys. Lett.* **17**, 553–558 (1992).
- [20] Jensen, J. & Macintosh, A. R. Rare Earth Magnetism: Structures and Excitations. (Clarendon, Oxford, 1991), p. 125.
- [21] Jensen, J. Theory of commensurable magnetic structures in holmium. *Phys. Rev. B* **54**, 4021–4032 (1996).
- [22] Baran, S., Duraj, R. & Szytula, A. Magnetocaloric effect and transition order in HoAl. *Acta*

- Phys. Polonica A* **127**, 815–817 (2015).
- [23] Moya, X., Kar-Narayan, S. & Mathur, N. D. Caloric materials near ferroic phase transitions. *Nat. Mater.* **13**, 439–450 (2014).
- [24] Yu, B., Liu, M., Egolf, P. W. & Kitanovski, A. A review of magnetic refrigerator and heat pump prototypes built before the year 2010. *Int. J. Refrigeration* **33**, 1029–1060 (2010).
- [25] Barclay J. A. & Steyert, W. A. (1982) Active magnetic regenerator. US Patent No. 4,332,135.
- [26] Hirano, N., Nagaya, S., Takahashi, M., Kuriyama, T., Ito, K. & Nomura, S. Development of magnetic refrigerator for room temperature application. *Adv. Cryog. Eng.* **47**, 1027–1034 (2002).
- [27] Utaki, T., Nagawa, T., Yamamoto, T., Kamiya, K. & Numazawa, T. Research on a Magnetic Refrigeration Cycle for Hydrogen Liquefaction. Cryocoolers 14, S. D. Miller & R. G. Ross Jr. (Eds.). Kluwer Academic/Prenum Publishers, pp. 645–653 (2007).
- [28] Hashimoto, T., Kuzuhara, T., Sahashi, M., Inomata, K., Tomokiyo, A. & Yayama, H. New application of complex magnetic materials to the magnetic refrigerant in an Ericsson magnetic refrigerator. *J. Appl. Phys.* **62**, 3873 (1987).
- [29] Behrendt, D. R., Legvold, S. & Spedding, F. H. Magnetic properties of dysprosium single crystals. *Phys. Rev.* **109**, 1544–1547 (1958).
- [30] Adachi, K., Matsui, M. & Kawai, M. Further investigations on magnetic properties of $\text{Co}(\text{S}_x\text{Se}_{1-x})_2$, ($0 \leq x \leq 1$). *J. Phys. Soc. Jpn.* **46**, 14741482 (1978).
- [31] Wada, H., Tanaka, K. & Tajiri, A. Magnetocaloric effect of $\text{Co}(\text{S}_x\text{Se}_{1-x})_2$. *J. Magn. Magn. Mater.* **290–291**, 706708 (2005).

End Notes

Acknowledgements

This work was supported by JSPS KAKENHI Grants (No. 17KK0099), and JST-Mirai Program Grant Number JPMJMI18A3, Japan.

Author contributions statement

N.T. and H.M. carried out the magnetization measurements and the thermal measurements, analyzed the data, and wrote the manuscript.

Competing interests

The authors declare no competing financial interests.

Corresponding authors

Correspondences to Noriki Terada and Hiroaki Mamiya.

Figure legends

FIG. 1: (a) Thermal variation of magnetic entropy change in typical magnetic fields (H) from $H/J = 0$ to $H/J = 5.0$ (where J is the ferromagnetic exchange constant) for ferromagnetic cases simulated in a previous theoretical study in Ref. [10]. The temperature is normalized to the Curie temperature T_C . (b) Schematic illustration of magnetic refrigeration (MR) cycle for the case of antiferromagnetic (AFM) material that shows a metamagnetic phase transition. Magnetic fields are represented by zero field $\mu_0 H = 0$ and finite fields $H_1 < H_2 < H_3 < H_4 < H_5$. For the AFM case in (b), the metamagnetic phase transition occurs over a small magnetic field change, $\Delta H = |H_2 - H_3|$, during the MR cycle $A' \rightarrow B' \rightarrow C' \rightarrow A'$.

FIG. 2: Magnetization curves along the hexagonal $[10\bar{1}0]$ direction at temperature from 2 K to 200 K in holmium single crystal.

FIG. 3: Magnetic entropy change ΔS_M as a function of temperature and magnetic field along the hexagonal $[10\bar{1}0]$ direction in holmium single crystal. The ΔS_M data are estimated by Eq. (1) with an integral range from $\mu_0 H_1 = 0$ T to $\mu_0 H_2 = \mu_0 H$, from the observed magnetization data.

FIG. 4: Magnetic field dependence of magnetic caloric efficiency, $-\Delta S_M / \Delta \mu_0 H$ ($\text{J kg}^{-1} \text{K}^{-1}$), for the temperature range from 20 K to 45 K in Ho. The magnetic field was applied along the hexagonal $[10\bar{1}0]$ direction. The $-\Delta S_M / \Delta \mu_0 H$ data are calculated by $-\Delta S_M$ divided by $\Delta \mu_0 H = 0.1$ T. The $-\Delta S_M$ is estimated by using Eq. (1) with the integration range from $\mu_0 H_1 = \mu_0 H$ to $\mu_0 H_2 = \mu_0 H + \Delta \mu_0 H$. For comparison between the present results and those for a typical magnetocaloric material, the efficiency for ferromagnetic HoAl₂ is also shown in the inset. The data for HoAl₂ were taken from a previous paper.[22]

5

FIG. 5: (a) Mapping of temperature change ΔT as a function of the bias magnetic field $\mu_0 H_0$ and temperature. (b) Typical heat cycles when the magnetic field rises from each $\mu_0 H_0$ by $\Delta \mu_0 H = 0.4$ T. The inset shows a magnification of the most efficient cycle for $\mu_0 H_0 = 0.5$ T and $T = 28.2$ K.

FIG. 6: (a) Schematic illustration of a cross section of Active Magnetic Regenerator (AMR) setup.[2, 26] The AMR constitutes a superconducting magnet, magnetocaloric materials, refrigerant space, and heat baths (low- and high-temperature sides). The magnetocaloric materials have different phase transition temperatures to obtain a large magnetocaloric effect at each position. (b) Possible setup of the proposed magnetic refrigeration (MR) system using a small change of magnetic field. The system constitutes pairs of permanent magnets, magnetocaloric material (holmium), refrigerant space, and heat baths. The set of permanent magnets generate a magnetic field gradient from 0.2 T to 1.2 T as a bias magnetic field $\mu_0 H_0$ to change the magnetic phase transition temperature of the magnetocaloric material Ho. The magnetic field change $\Delta \mu_0 H$ is realized by mechanically moving the magnetocaloric material in the horizontal direction in this figure. (c) The MR cycle of the proposed AMR system. The gray color in the magnetic field and temperature graphs denotes the bias field $\mu_0 H_0$ and the original temperature T_0 , respectively. The colored areas in the graphs are the changes in the magnetic field $\Delta \mu_0 H$ and temperature ΔT .

Reviewers' Comments:

Reviewer #2:

Remarks to the Author:

- All answers and explanations to the questions that I raised in my first report are satisfactory.
- The added comments are satisfactory as well.

Conclusion: I have no longer objection to the publication of this work.

Reviewer #3:

Remarks to the Author:

The authors have satisfactorily addressed the concerns which I raised in the previous version. I can accept this paper for publication.